# The Impact of Limited Previous Motor Experience on Action Possibility Judgments in People with Spinal Muscle Atrophy

**DOI:** 10.3390/brainsci13091256

**Published:** 2023-08-29

**Authors:** Sarvenaz Heirani Moghaddam, Dilara Sen, Megan Carson, Robert Mackowiak, Rachel Markley, Gerome Aleandro Manson

**Affiliations:** 1School of Kinesiology and Health Studies, Queen’s University, Kingston, ON K7L 3N6, Canada; 2Houston Methodist Research Institute, Houston, TX 77030, USA

**Keywords:** sensorimotor integration, spinal muscle atrophy, cognition, Fitts’s law, motor, sensory

## Abstract

Previous studies have shown that people with limited motor capabilities may rely on previous motor experience when making action possibility judgments for others. In the present study, we examined if having limited previous motor experience, as a consequence of spinal muscle atrophy (SMA), alters action possibility judgments. Participants with SMA and neurologically healthy (NH) sex- and age-matched controls performed a perceptual-motor judgment task using the Fitts’s law paradigm. Participants observed apparent motion videos of reciprocal aiming movements with varying levels of difficulty. For each movement, participants predicted the shortest movement time (MT) at which a neurologically healthy young adult could accurately perform the task. Participants with SMA predicted significantly longer MTs compared to controls; however, the predicted MTs of both SMA and NH participants exhibited a Fitts’s law relationship (i.e., the predicted MTs significantly increased as movement difficulty increased). Overall, these results provide evidence that participants with SMA who have limited, or no motor experience may make more conservative action possibility judgments for others. Critically, our finding that the pattern of action possibility judgments was not different between SMA and NH groups suggests that limited previous motor experience may not completely impair action possibility judgments.

## 1. Introduction

Making accurate predictions about our own actions and the actions of other individuals is important for effectively interacting with others and our environment. For example, prior to passing an object to a small child, we need to judge if the child can safely hold and maneuver the object. Previous studies have suggested that such action possibility judgments rely on a simulation process that utilizes linked action and perception networks [1,2,3,4]. Specifically, when asked to make a judgment about another person’s action, the actor first stimulates the action and forms a prediction about their own performance. When predicting for another person, the actor uses their own estimate as a basis, and either adds or subtracts a correction factor based on their perception of the other individual’s characteristics (e.g., [5]).

Evidence for the use of simulations in action possibility judgments has emerged from studies examining the ability to predict MT based on movement difficulty [2,3,5]. Specifically, these studies used a Fitts’s Law paradigm that characterizes the relationship between MT and movement difficulty for reciprocal upper-limb reaching movements [2,3,5,6]. The Fitts’s Law equation which captures the relationship between MT and movement difficulty is:MT = a + b (log2[2A/W]),(1)
where “a” and “b” are constants that relate one’s baseline MT (y-intercept) and the change in MT for a given change in movement difficulty (slope of the regression line), respectively. To quantify the index of difficulty (ID) of the movement in bits, the following equation is used [7]: ID = log2[2A/W],(2)

The ID is therefore a function of the target width (W) and the movement amplitude which is defined as the distance between the two targets (A). In general, as the width of the target decreases and/or movement amplitude increases, ID, and as a result, MT also increases.

Using the reciprocal aiming Fitts’s Law paradigm, Grosjean et al. (2007) [2] investigated if participants’ action possibility judgments about upper-limb movements followed a Fitts’s Law relationship. In this study, participants observed displays of an arm moving between two targets and estimated the MT. As in the classic Fitts’s law experiment, the distance between the targets as well as the width of the targets were varied. The authors found that the shortest estimated MTs corresponded with the lowest ID and as ID increased, MT increased linearly, showing evidence of a Fitts’s Law relationship. The authors concluded that the observation of a Fitts’s law relationship provided evidence that participants were simulating their reach performance before making action possibility judgments.

The finding suggesting that action possibility judgments involve a simulation of the movement being judged supports the idea that the motor system plays a crucial role in action-perception. This is supported by experiments showing that judgments about others’ actions are affected by recent motor experience and current body state. For instance, Chandrasekharan et al. (2012) [1] observed that participants’ predicted MTs were shorter after they had performed the Fitts’s law task. Conversely, the authors also found that predicted MTs were longer when participants performed action possibility judgments while wearing a weight on their wrist (altered body state). These findings demonstrate that recent motor task experience and altered body state can modulate the simulation process and influence the magnitude of the action possibility judgments. 

Extending the idea that the current body state can influence action predictions, Manson et al., 2014 [3] examined if changes in motor capabilities as a result of neurological injury could influence one’s ability to perform action possibility judgments. In their study, participants with limited upper-limb function due to cervical spinal cord injury (cervical SCI) and control participants with typical upper limb function performed action possibility judgments using Fitts’s law paradigm. Critically, participants made judgments for both themselves and for neurologically healthy young adults. The authors found that participants with cervical SCI had significantly longer predicted MTs compared to controls when predicting for themselves; however, there were no differences between participants with cervical SCI and the control group when predicting for the healthy young adult. Furthermore, the authors found that the predicted MTs for both groups followed a Fitts’s law relationship in both judgment tasks. 

The authors forwarded two possible hypotheses for their results. First, individuals with cervical SCI were able to form action possibility judgments for the healthy young adult by utilizing intact central action-perception networks that were developed pre-injury. This hypothesis is based on findings that perception of other’s actions is heavily influenced by previous motor experience (e.g., [8,9]). The second hypothesis was that individuals with cervical SCI first simulated their own perceived performance using their unimpaired central action-perception networks, and then modified action possibility judgments based on the perceived differences between themselves and the healthy young adult. This latter, and preferred hypothesis, was based on the finding that the pattern of action possibility judgments (i.e., the slope of the relationship between ID and MT) was not different between the self and other action possibility judgment tasks. 

Given the results from Manson et al., (2014) [3], it could be possible that people with acquired SCI used their pre-injury networks to make accurate action possibility judgments. Thus, it is currently unclear whether severely limited motor experience from birth affects action possibility judgments. The goal of the present study was to examine if prior motor experience is critical for the action simulation process used to predict the other’s actions. To address this question, we selected participants with spinal muscle atrophy (SMA). SMA is a genetic neuromuscular disease that primarily affects motor neurons in the spinal cord leading to muscle loss and weakness in the upper and lower-limbs (see: [10]). Symptoms appear from 4–18 months of age and last throughout adulthood. Importantly, because of the early onset and nature of the pathology, people with SMA have largely intact central processing networks, but severely limited motor experience from birth. 

The action possibility judgments of participants with SMA were compared to neurologically-healthy age, and sex-matched controls using the Fitts’s Law paradigm. It was hypothesized that, if previous motor experience forms the basis of action possibility judgments, then both the predicted MTs and pattern of action possibility judgments would be different in participants with SMA than in controls. In contrast, if action possibility judgments are based on a more cognitive representation of one’s performance (whether it be real or perceived), then we expect that only the predicted MTs and not the pattern of action possibility judgments would be different between participants with SMA and controls. This second hypothesis considers the importance of prior motor experience (and recent practice) in the modification of the accuracy of judgments, but not the pattern of predictions (see [5,6]). Overall, the results of the present study support the latter hypothesis and suggest that prior motor experience may lead to more accurate judgments but may not impact the overall pattern of predictions. 

## 2. Materials and Methods

### 2.1. Participants

Fifteen (15) participants with Spinal Muscle Atrophy (SMA group, nine Females, Age: Mean (M) = 41, Standard Deviation (SD) = 12; nine Right-handed; see Table 1 for detailed demographics) took part in this experiment. Following data collection for the SMA group, 15 age- and sex-matched neurologically healthy participants (NH group; 13 Right-handed participants) were recruited. All subjects gave their informed consent for inclusion before they participated in the study. The study was conducted in accordance with the Declaration of Helsinki, and the protocol was approved by the Queen’s University General Research Ethics Board (GREB), GSKHS365-20 (Project identification code). Participants were compensated with a $20 Gift card for their time. 

### 2.2. Apparatus, Stimuli, Task

The experimental protocol was completed online. Prior to the start of the experiment, participants were contacted by experimenters using Zoom video conferencing software (Version number: 5.515.7 (21404), Zoom Video Communications Inc. Kingston, Ontario, Canada, 2021). The experimenter remained on the videoconference call with the participant throughout the duration of the experiment and all experimental tasks were completed on the participant’s own computer. After informed consent was provided, participants were presented with standardized instructions about the experiment (details below). Following the instructions, participants completed the experimental task. A custom PsytoolKit program was used to launch the experimental task and to collect data [11,12]. 

The action possibility judgment task in this experiment was similar to previous studies (see [2,3,6]). In brief, nine movement stimuli were presented to the participants. Each stimulus consisted of a white background paper with two black strips (15 cm in height) that served as targets for reciprocal aiming movements. The target widths and the amplitude between the two targets were varied to yield movements with an index of difficulty of either 2, 3, or 4 bits (see Figure 1A,B). 

Apparent motion was generated by alternating two photos of an actor (i.e., a model participant) sitting with a finger on the right target and a finger on the left target for each of the nine movement stimuli (see Figure 1C). The same pairs of pictures were displayed throughout a single trial so that the index of difficulty was consistent within a trial. The time between the presentation of the two pictures (the stimulus onset asynchrony [SOA]) served as the predicted MT for the judgment tasks.

Participants were instructed to select the fastest possible MT that they believed the person in the video (PIV) could achieve while maintaining accuracy. Participants were told that the PIV was a neurologically healthy 26-year-old male (see [3]). To begin a trial, participants moved the cursor to a yellow box located at the bottom left of their screen to display the first task condition. The SOA at the beginning of each trial was randomly selected ranging from 30–890 ms. Participants were instructed to use the up and down arrow keys on their keyboard to adjust the speed of the apparent motion video (Figure 1B). The up and down keys either increased or decreased the time delay between the first and the second image being shown by 20 ms. Some participants in the SMA group (N = 3) used voice commands “up” and “down” with a voice control device to complete the experiment. Once participants selected their MT, they hovered their mouse over a green square on the right side of the screen. The experiment continued when participants moved their cursor back to the yellow box to start the next trial. Each participant completed a total of 45 trials such that each of the nine combinations was used five times. The presentation order of the stimuli was randomized. 

### 2.3. Spinal Muscle Atrophy Health Index (SMAHI)

Following the experiment, participants in the SMA group completed a modified version of the “Spinal Muscular Atrophy Health Index” to examine participants’ upper-arm function (SMAHI; [13]; see Appendix A). The modified SMAHI asked participants to report arm mobility, hand mobility, and their use of assistive technology (i.e., wheelchair, walker, cane). Based on the Likert scale from the SMAHI, a factor scale of “0–5” was created, where “I don’t experience this” corresponded to a score of “0” and “It affects my life severely” corresponded to a score of “5”. In the case of the last question which asked: “Which of the following would best describe how you get around?” where multiple answers were possible, the more assistance one needed from an outside device or technology, the higher the score (e.g., cane = 1, motorized wheelchair = 5). Each participant received a score out of 5 on each question. The average score was then calculated for each participant. The average score on the SMAHI can be found in Table 1 (also see Appendix A for an example of a participant’s SMAHI answers).

### 2.4. Data Reduction and Analysis

One SMA participant was excluded from further analyses because they completed the study in an environment that was not conducive to the task (in a vehicle). Their matched control was also removed from the analysis. Therefore, the analysis below was performed on a total of 28 participants, 14 participants in the SMA group and 14 participants in the NH group (see Table 1 for demographic information).

A custom MATLAB script was used to remove outliers (Mathworks, 2022). First, a hard criterion was set to remove any MT less than 100 ms and MT over 1000 ms. This hard criterion resulted in the removal of 12 trials in total (nine trials from the SMA group and three trials from the NH group, 0.0089% of the overall number of trials). Following the removal of trials based on the hard criteria, trials that were 2.5 standard deviations above or below the participant’s mean for a given stimulus were also removed. This resulted in a further removal of eight trials in total (seven from the SMA group and one from the NH group, 0.0059% of the overall number of trials). In total, 20 trials were removed from the entire data set (0.016% of the overall number of trials).

A series of planned comparisons were conducted to test the experimental hypotheses. Linear regressions between the MTs for each individual participant in the SMA and NH groups were conducted to examine the relationship between predicted MT and ID (2, 3, and 4 bits). A regression analysis between MT and ID was also completed using participant-level averages to examine group-level relationships between NH and SMA participants. The purpose of these analyses was to determine if the MTs in each of the IDs conformed to Fitts’s Law.

A second analysis was performed to examine if the regression lines for the Fitts’s Law equation were different between groups. The y-intercepts and slopes obtained from the individual regression lines for participants in the SMA and NH groups were compared using paired samples *t*-tests. This analysis was used in previous studies to examine if there were systematic differences in how participants adjusted their action possibility judgments based on other’s characteristics (see [1]).

To test whether the groups differed in predicted MTs across different IDs mean predicted MTs averaged across the different combinations of distance and width for a given ID for everyone on the action possibility judgment task were submitted to a 2 Group (SMA, NH) × 3 ID (2, 3, 4) mixed ANOVA with Group as a between-subject factor and ID as the within-subject factor. When the assumption of Sphericity was violated, the Huynd-Feldt correction was used to correct the degrees of freedom.

### 2.5. SMA Subgroup Analysis

As a supplementary analysis, the SMA group was divided into a “SMA: no motor function” group and a “SMA: some motor function” group based on participants’ responses to the SMAHI (see Table 1 for details). Specifically, participants with SMAHI scores of 4.5–5 were classified into the SMA: no motor function group while participants with SMAHI scores of <4.5 were classified into the SMA: some motor function group. A series of linear regressions between the mean predicted MTs and ID were performed for both subgroups. The purpose of these analyses was to determine if the relationship between ID and predicted MT was affected by the severity of motor impairment due to SMA. Further, to examine differences between the subgroups of SMA, we completed a 2 Group (SMA: No Motor Function and SMA: Some Motor Function) × 3 ID (2, 3, 4) mixed ANOVA with Group as a between-subject factor and ID as repeated measures factor.

## 3. Results

### 3.1. Fitts’s Law at the Participant Level

#### 3.1.1. Individual Regressions

We visually inspected the relationship between the predicted MT and the three IDs for each individual participant (see Figure 2). Individual regression analyses revealed that 11 out of 14 participants in the SMA group and 11 out of 14 participants in the NH group showed evidence of Fitts’s law.

#### 3.1.2. Comparisons of y-Intercepts and Slopes

The comparison of the y-intercepts and slopes between the SMA and NH groups using paired samples *t*-test revealed a significant difference of y-intercept between groups: SMA: M = 276 ms, SD = 148; NH group: M = 176, SD = 81; *t*(13) = 2.052, *p* = 0.0351) but no significant differences of slope between groups (Slope: SMA: M = 75 ms, SD = 39; NH: M = 71, SD = 37; (*t*(13) = 1.121, *p* = 0.809).

### 3.2. Fitts’s Law at the Group Level

#### 3.2.1. Movement Time as a Function of Index of Difficulty

The results of the linear regression analyses revealed that predicted MTs for the SMA and NH groups conformed to Fitts’s Law (see Figure 3). The results of the regression analyses are as follows: SMA group, MT = 275.7 + 74.7(ID), *R*^2^ = 0.225, *p* < 0.001; NH group, MT = 175.66 + 71.05(ID), *R*^2^ = 0.2748, *p* < 0.001).

#### 3.2.2. Movement Time Predictions

The mixed ANOVA analysis of predicted MTs between SMA and NH groups revealed a main effect of Group (*F*(1, 26) = 8.665, *p* = 0.007, η^2^ = 0.227) and ID (*F*(1.5, 38.88) = 79.583, *p* < 0.001, η^2^ = 0.288) but no Group x ID interaction (*F*(1.5, 38.88) = 0.097, *p* = 0.853, η^2^ = 0.0004; see Figure 4). 

### 3.3. Fitts’s Law at the Subgroup Level

#### SMA Subgroup Regression and ANOVA

The SMA: Some Motor Function subgroup followed Fitts’s Law (MT = 277.1 + 77.7(ID), *R*^2^ = 0.24, *p* = 0.004949) but the SMA: No Motor Function subgroup did not follow a Fitts’s Law relationship (MT = 256.6 + 71.8(ID), *R*^2^ = 0.136, *p* = 0.0968) (see Figure 5). 

A 2 group by 3 ID mixed ANOVA was performed on predicted MTs on predicted MTs between SMA: No Motor Function and SMA: Some Motor Function subgroups and revealed a main effect of ID (*F*(1.25, 15) = 30.022, *p* = 0.606, η^2^ = 0.020) but no main effect of group (*F*(1, 12) = 0.281, *p* < 0.001, η^2^ = 0.232) and no interaction between Group × ID (*F*(1.35, 15) = 0.233, *p* = 0.69, η^2^ = 0.002; see Figure 6).

## 4. Discussion

The purpose of this study was to investigate the impact of limited motor experience on the perception of others’ actions. Participants with SMA and NH participants predicted the shortest possible time that a healthy young adult male could perform a reciprocal aiming task. It was found that participants with SMA had significantly longer predicted MTs than NH controls. Critically, analyses of the regression lines at the group level revealed that the differences in MT predictions were driven by higher baseline predictions (i.e., the y-intercepts were higher in the SMA group) rather than differences in the pattern of predictions (i.e., the slope of the regression lines). Thus, the relationship between movement difficulty and predicted MT was not different between participants with SMA and NH controls. Further analyses revealed that there were no differences in predicted MTs between participants with SMA with some motor function and those with no motor function as categorized by the SMAHI. This result suggests that action-perception was not reliably altered by the severity of SMA symptoms. 

Overall, our findings provide evidence that central action-perception networks may still be functional in people with limited previous motor experience. Furthermore, simulations using these networks were likely still employed by participants with SMA when making action possibility judgments for others. Finally, our results suggest that although action predictions are informed by previous motor experience, this information may be more important for the fine-tuning of judgments. The following discussion will focus on the impact of limited motor experience on the development of action-perception networks and the formation of action possibility judgments.

In contrast to previous studies (e.g., [3,14]), participants with limited motor capabilities, as a result of SMA, predicted significantly longer MTs than control participants when making action possibility judgments for a neurologically healthy individual. One possible explanation for this finding is that the limited motor experience of participants with SMA could have impaired the development of the action-perception networks that are used to form predictions for others. This hypothesis is supported by the findings of Manson et al., (2014) [3] who demonstrated that participants with limited upper-limb function (as a result of cervical SCI), but previous motor experience (i.e., full arm function prior to injury) were able to adjust their predictions when forming action possibility judgments for healthy individuals. 

The importance of previous motor experience in the formation of action-perception networks has also received support from both behavioral and neuroimaging experiments (see: [8,9,15,16]). For example, Stapel and colleagues [16] used gaze behavior to determine if children (and adults) with different levels of motor experience could predict the timing and trajectory of different types of locomotion (e.g., crawling, walking), and object translation. The authors found that participants were better at forming predictions for movements that were in participants’ motor repertoires. For example, infants who were proficient crawlers, but not proficient walkers, were better at predicting the timing of crawling than walking (see [16]).

Although the abovementioned studies provide evidence that previous motor experience likely contributed to the differences in predicted MTs between SMA and control groups, two of our findings suggest that previous motor experience may not be necessary to form reasonable action possibility judgments. First, the predicted MTs of both the SMA and control groups followed a Fitts’s Law relationship where predicted MTs were significantly positively correlated with movement difficulty. Second, comparisons of the slopes of the regression lines between groups of participants revealed that the pattern of predictions, that is, the impact of increases in difficulty on the changes in predicted MTs was no different between groups. 

The finding that predicted MTs linearly increased with ID suggests that functional action-perception networks could have been developed in participants with SMA even though they had limited motor experience. Seminal studies that employed the Fitts’s Law paradigm have argued that the presence of a significant correlation between MT and ID (i.e., the Fitts’s Law relationship) indicates that linked action-perception networks were engaged for action possibility judgments (see: [2,5,6,17,18]). Furthermore, the absence of a Fitts’s law relationship has been associated with dysfunction in central action-perception networks (see [17,18]). Eskanazi and colleagues [18] used Fitts’s paradigm to investigate if predicted MTs would be impaired in a participant with a frontal brain lesion (i.e., a stroke-induced lesion that affected the left inferior, middle, and superior frontal gyri, see [17]). The authors found that the participant’s predicted MTs were correlated with movement amplitude, but not scale to the movement’s ID. The absence of a Fitts’s Law relationship provided evidence that the impaired brain regions, which have also been associated with action-perception in Fitts’s task and other contexts, were not engaged during predictions (see: [18,19]). Considering the aforementioned findings, the presence of a Fitts’s Law relationship in the current study lends support to the idea that participants with SMA used functional cortical action-perception networks to make predictions for others.

The hypothesis that functional action-perception networks were employed by participants with SMA to make action possibility judgments is also supported by the finding that there were no reliable differences in the pattern of predictions (i.e., the slopes of the regression lines) between participants with SMA and NH controls. Thus, for both groups, an increase in movement difficulty resulted in a similar increase in predicted MT (see Figure 3). This pattern of results is similar to those described by Manson et al., (2014) [3] where the authors found that there were no differences in the slopes of the regression lines when participants with cervical-SCI and controls predicted MTs for themselves and a neurologically healthy adult. Critically, there were also no differences in slopes between the cervical SCI and control group when predicting for the neurologically healthy adult. Based on these results, the authors concluded that both groups of participants engaged in a simulation process using linked action-perception networks when predicting the actions of others. Predicted MTs were therefore hypothesized to be derived based on an adjustment that considered the perceived differences between one’s own capabilities and the perceived capabilities of the other actor (see also [1]). Based on the results of the present study, it is hypothesized that participants with limited motor experience as a result of SMA engaged in an action-simulation process that was similar to NH controls. That is, participants with SMA simulated the movement using intact central simulation networks and then adjusted their judgment based on the perceived differences in capabilities between their simulation and the characteristics of the actor. 

The idea that participants with limited previous motor experience could simulate actions that they could not perform themselves contrasts with literature presented earlier about the importance of motor capabilities in the formation of action-perception networks [8,9,15,16]. However, beyond the results of the present study, further support for this hypothesis can be drawn from studies examining action perception in children with cerebral palsy (e.g., [20] see also [21] for a brief overview). For example, Dinomais and colleagues [20] found that the temporo-frontal and parieto-occipital brain regions that are active during action observation in adults were similarly activated when children and adolescents with congenital unilateral hemiparesis observed arm movements. Importantly, activation in these action-perception networks was present when children viewed movements of limbs that corresponded to both their unimpaired and impaired limbs. This finding aligns well with studies showing that people born without limbs (congenital amputees) also engage in motor, and more widespread visual networks when observing other’s limb movements and predicting the actions of others [22,23,24]. Although no studies have examined action observation in SMA, previous work has shown that visuospatial cognitive function is preserved, and can be further enhanced through early locomotor experience, in children with SMA (see [25,26]). Although further neuroimaging work is needed, it could be that participants with SMA employed central action perception networks to simulate actions and form action possibility judgments. 

Although our results suggest that previous motor experience may not be critical for the formation of action possibility judgments (see also [23,24]). Our data do suggest that motor experience could be used to refine the accuracy of action possibility judgments. In studies by Welsh et al., 2013 [5], and Wong et al., 2013 [6], the authors found that participants’ predictions about their own MTs were more closely matched to their actual MTs after physically performing the movement themselves. The idea that previous motor experience can be used to refine judgments could also partially account for the contrasting results between our study and Manson et al., 2014 [3]. In their study, participants performed the task prior to completing both self and other action possibility judgments. Although participants with cervical SCI in Manson et al., 2014 [3] only had partial limb function and performed the task slower than control participants, perhaps recent task experience allowed them to refine their judgment for others. If this is the case, the results of the present study could be initial evidence that passive movement therapy (perhaps with robotic guidance see: [27,28]) could impact action-perception in people with limited motor capabilities due to SMA. 

It is also important to mention that the perception of action possibility judgments can be influenced by factors beyond previous motor experience. Differences in cognitive processing and/or task familiarity could have contributed to the observed differences in predicted MTs [29,30]. For example, Zelaznik and Forney [31], found that perception of Fitts’s task difficulty (target width) was only related to motor performance if participants received a score after they performed the task. This finding indicates that knowledge of performance, in addition to motor experience is important for the accuracy of perceptual judgments (also see [32]). Furthermore, in the current study, it is possible that familiarity with speed accuracy tradeoffs could impact perceptual judgments. Although no studies have investigated whether repeated observation can improve perception in the absence of motor experience for upper-limb reaching tasks, it is possible that repeated action observation could improve prediction ability. This hypothesis is based on findings that action-observation and motor imagery facilitate motor skill acquisition (for a review see [33]).

One limitation of the present study is that we did not collect data about the specific diagnosis of participants (i.e., the type of SMA), or if they were on medication at the time of the experiment. Although the SMAHI and pre-experiment screening questions ensured they met the criteria for limited previous motor experience, it is unknown if being on medication could have influenced action possibility judgments. Finally, another limitation of the current study is the exclusion of the self-judgment task (see [3,14,17]). While the inclusion of this condition could have provided concrete data on how participants with SMA perceived their own ability, initial pilot tests revealed that participants with SMA consistently chose the slowest possible times for every ID when asked to estimate “how fast they could perform the action possibility judgment task”.

## 5. Conclusions

This study was designed to investigate the impact of limited previous motor experience on the perception of others’ actions. Our findings indicate that participants with limited previous motor experience due to SMA predicted significantly longer MTs than NH controls when performing action possibility judgments for a reciprocal upper limb aiming task. Although the predicted MTs were longer, the pattern of predictions scaled to the difficulty of the movements and was not different between people with SMA and NH controls. These findings challenge the notion that previous motor experience is required to form action possibility judgments and support the idea that functional central action-perception networks are present in individuals with limited motor experience. 

## Figures and Tables

**Figure 1 brainsci-13-01256-f001:**
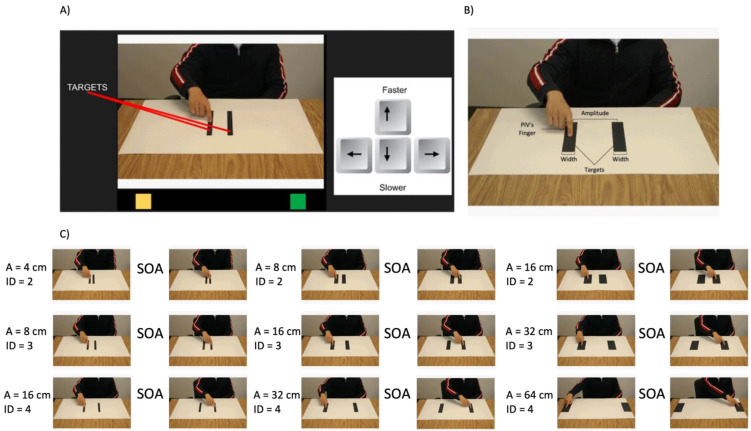
(**A**) Depicts the stimuli used for the experiment, including the Person in the Video (PIV), yellow and green squares. The image also shows an arrow keyboard layout that served as the control scheme used to adjust predicted MTs. The green square indicated the end of a trial such that when the participant hovered their mouse over the green square, a trial ended. The yellow square indicated the start of a trial such that when the participants’ cursor hovered over the yellow square, a trial started. (**B**) Details included in the apparent motion videos. The two targets included in apparent motion videos either had different widths or amplitudes. In all apparent motion videos, PIV’s finger moved between the two black targets placed on a white poster. (**C**) Apparent motion videos with varying Index of Difficulty (ID): 2, 3 and 4. Within each ID, distance between the targets (A: Amplitude) and the width of the targets was manipulated. Images on the left indicate the PIV’s finger on the right target followed by an arrow indicating the stimulus onset asynchrony (SOA) to the second image with the PIV’s finger on the left target.

**Figure 2 brainsci-13-01256-f002:**
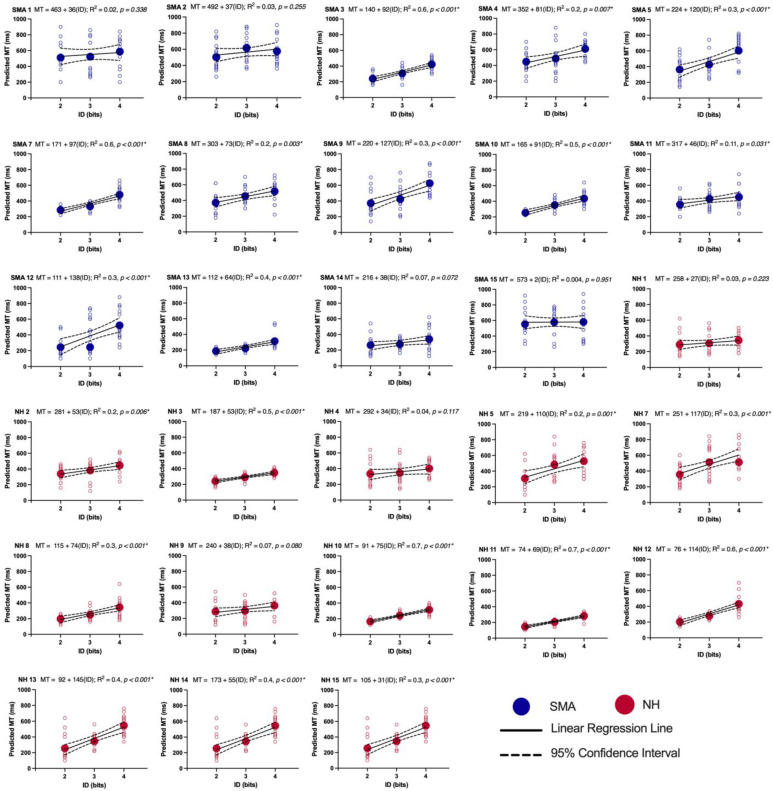
Individual Fitts’s plots. Predicted MT (ms) across trials for each participant’s individual trials (small circles) and average predicted MT for ID 2, 3 and 4 (big circles). The equation for linear regressions, the *R^2^* and *p*-values are located on top of each graph. Asterisk (*) indicate Fitts’s Law.

**Figure 3 brainsci-13-01256-f003:**
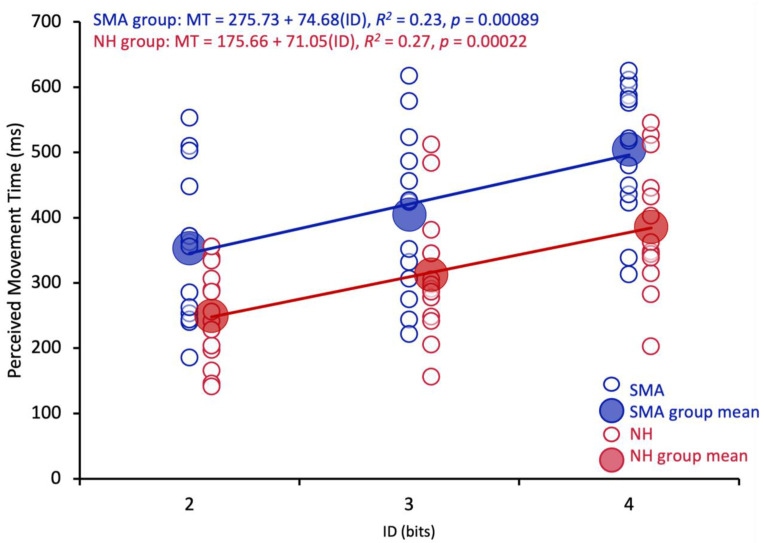
Average predicted MT (ms) across trials for each participant (small circles) and overall group average (big circles) for the SMA group (blue) and NH group (red) for ID 2, 3 and 4. SMA: ID is index of difficulty (bits). SMA group: MT = 275.7 + 74.7(ID); NH group: MT = 175.66 + 72.05(ID).

**Figure 4 brainsci-13-01256-f004:**
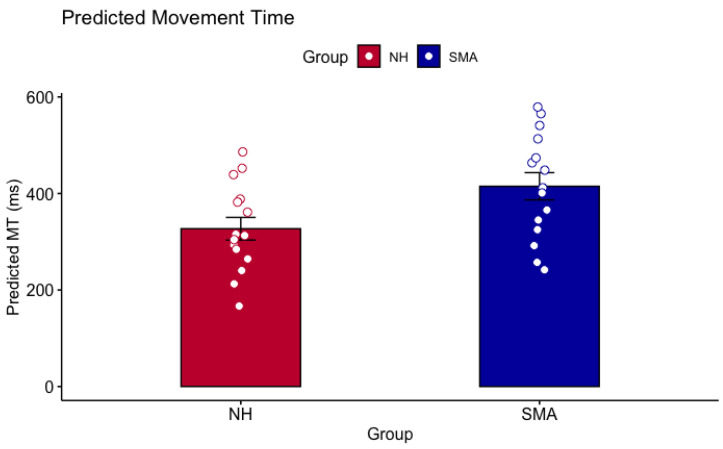
Overall predicted MT (ms) for SMA group (blue) and NH group (red). Error bars are the standard error of the mean. Small unfilled circles are individual participant data. The ANOVA revealed a significant group difference where the predicted MTs of the SMA group were significantly higher than the predicted MTs of the NH group.

**Figure 5 brainsci-13-01256-f005:**
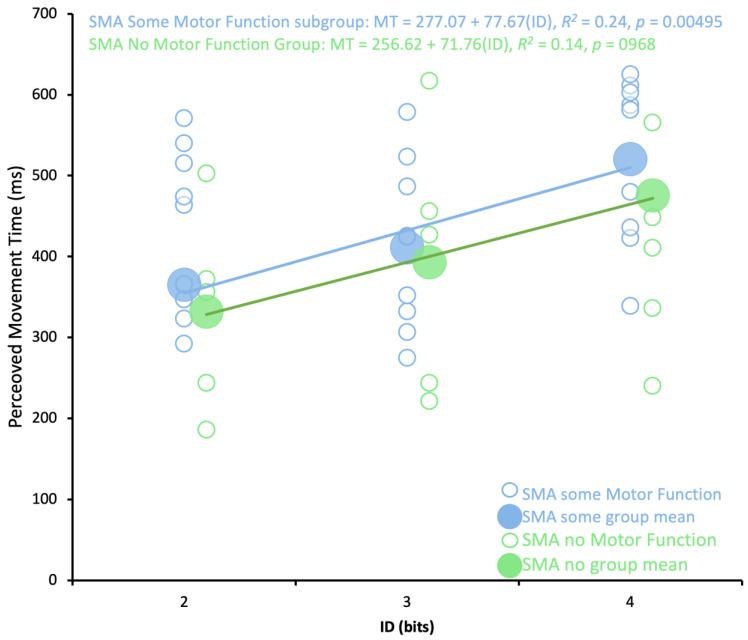
SMA subgroups. Average predicted MT (ms) across trials for each participant (small circles) and overall group average for ID 2, 3 and 4 (big circles). SMA: Some Motor Function (light blue) and SMA: No Motor Function (green). ID is index of difficulty (bits).

**Figure 6 brainsci-13-01256-f006:**
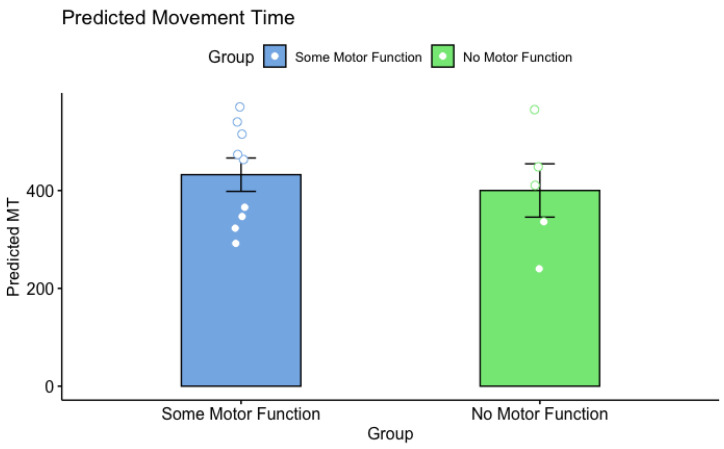
Overall predicted MT (ms) for SMA: No Motor Function (Green) and SMA: Some Motor Function subgroups (Blue). Error bars are standard error of the mean. Small unfilled circles are individual participant data. The ANOVA revealed no significant differences between the sub-groups.

**Table 1 brainsci-13-01256-t001:** The demographic data obtained for each participant in each group. The Edinburgh handedness inventory was used to determine participants handedness by asking them about which hand they use to perform daily activities such as writing, drawing, throwing, using a toothbrush and a broom. A modified version of the Spinal Muscular Atrophy Health Index (SMAHI) was completed only by the SMA group to examine participants’ arm mobility, hand mobility, and their use of assistive technology (i.e., wheelchair, walker, cane). Each participant received a score out of 5 on each question. The average score was then calculated for each participant and was reported in this table.

Group	Age	Gender	Handedness	SMAHI Score
Spinal Muscle Atrophy (SMA)	27	Female	None	3.4
	58	Female	None	5.0
	21	Female	Right	3.6
	57	Male	Right	3.6
	48	Female	Right	3.9
	34	Male	Right	3.0
	33	Male	None	5.0
	48	Female	Right	1.6
	54	Male	None	4.5
	37	Female	Left	5.0
	43	Female	Right	5.0
	56	Male	None	3.9
	33	Male	Right	3.2
	29	Male	Right	1.7
Neurologically Healthy (NH)	27	Female	Right	
	58	Female	Right	
	21	Female	Right	
	57	Male	Right	
	48	Female	Right	
	34	Male	Right	
	33	Male	Left	
	48	Female	Right	
	54	Male	Right	
	37	Female	Right	
	56	Male	Right	
	43	Female	Left	
	33	Male	Right	
	29	Male	Right	

## Data Availability

Data will be available upon request.

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
