# Peer review of "The Impact of Limited Previous Motor Experience on Action Possibility Judgments in People with Spinal Muscle Atrophy"

_brainsci, 2023, doi:10.3390/brainsci13091256_

Round 1
Reviewer 1 Report
The authors present an interesting and innovative evaluation of action possibility judgments of patients living with Spino Muscular Atrophy (SMA). Their findings report that prior motor experience could lead to more accurate action judgments without impacting the overall pattern of predictions. these findings indicated that central action-perception networks may still be functional in people with limited previous motor experience.
There are a few comments the authors need to address:
1. The authors mentioned the lack of information regarding the type of SMA diagnosed in SMA patients as well as their medication regimen as a limitation of their study. However, consideration of SMAHI scores to determine the extent of their disability is not sufficient. The authors did not provide the classification criteria for SMA subgroup analysis. Further clarification is required.
2. Figure 1A legend has a typo. the period (.) is seen twice.
3. Was the experiment conducted in person? How were they monitored?
4. Was the person in video (PIV) a participant? That was not very clear.
5. The font size in the figure is too small and cannot be read. Were high-resolution images created? if that is not possible, please increase the font to make it more readable.
6. Please list the color classification in Figure 2 legend. Again, the font is too small. Please adjust that.
Author Response
Dear Dr. Murphy,
We would like to thank the reviewers for their time and thoughtful comments. Based on the reviewers’ comments, we have made edits to the figures, the introduction, methods, and discussion sections of the manuscript and believe we have a stronger article because of these suggestions.
Below we have detailed our responses to each of the reviewer comments. The reviewer comments are listed in Italics, and our responses are listed in red. Where applicable we have also listed the page, and line numbers corresponding to the changes in the manuscript. The changes in the manuscript have also been documented using tracked changes. We have submitted a tracked changes version, changes in bold version and a clean version of the revised manuscript.
Sincerely,
Sarvenaz Heirani Moghaddam
Reviewer 1
The authors present an interesting and innovative evaluation of action possibility judgments of patients living with Spino Muscular Atrophy (SMA). Their findings report that prior motor experience could lead to more accurate action judgments without impacting the overall pattern of predictions. these findings indicated that central action-perception networks may still be functional in people with limited previous motor experience.
Response: Thank you to the reviewer for their supportive comments.
There are a few comments the authors need to address:
- The authors mentioned the lack of information regarding the type of SMA diagnosed in SMA patients as well as their medication regimen as a limitation of their study. However, consideration of SMAHI scores to determine the extent of their disability is not sufficient. The authors did not provide the classification criteria for SMA subgroup analysis. Further clarification is required.
Response: Thank you for the opportunity to clarify our sub-group analysis. Considering this point has helped us clarify the manuscript greatly. We did not use participant information about their sub-type for the subgroup analysis as we believe that most participants with SMA included in our study would’ve been sub-types II and III. We conducted our analyses using the information they provided about their upper-limb motor experience, which was derived from their responses to the modified SMAHI questionnaire. We believe this is a more relevant measure to classify participants in our study as it directly relates to the functional abilities associated with the experimental task.
To avoid further confusion about the subgroup analysis, we have also removed the descriptions of SMA sub-types from the introduction. Further, we have now added how we made the subgroup classification by saying the following on page 6, line 233:
“Specifically, participants with SMAHI scores of 4.5-5 were classified into the SMA: no motor function group while participants with SMAHI scores of < 4.5 were classified into the SMA: some motor function group.”
- Figure 1A legend has a typo. the period (.) is seen twice.
Response: Thank you for this comment, we have now fixed this issue.
- Was the experiment conducted in person? How were they monitored?
Response: Thank you for bringing up this point of clarification. This experiment was conducted online, and the experimenters were on a video call with the participants as they completed the task. The nature of the experiment has been made more explicit on pages 3, and line 133 by saying:
“The experimental protocol was completed online.”
- Was the person in video (PIV) a participant? That was not very clear.
Response: Thank you for bringing up this point of clarification. The images of the person in the video (PIV) were from an actor (i.e., not an experimenter or participant in the present study). We see where this point of confusion came from (the description of apparent motion and the description in figure 1) and we have modified the wording in the methods to make this clear. See page 4, lines 148 & on Page 5, line 170.
- The font size in the figure is too small and cannot be read. Were high-resolution images created? if that is not possible, please increase the font to make it more readable.
- Please list the color classification in Figure 2 legend. Again, the font is too small. Please adjust that.
Response: Thank you for the suggestions, we agree and we have now increased the font and/or used high resolution images for all of our graphical figures.
Reviewer 2 Report
Dear Authors,
I have carefully reviewed your manuscript and would like to offer a few minor comments for your consideration.
# Remove references from the abstract (e.g., see Fitts, 1954).
# Introduction: It Is not clear the rational for including individuals with Spinal Muscle Atrophy (SMA).
# The table: The demographic data obtained for each participant in each group should be moved in to the result section.
# Authors should discuss that the perception of action possibility judgments could be influenced by factors other than motor experience, such as cognitive processing differences or familiarity with the task. These factors might not have been adequately controlled for.
Author Response
Dear Dr. Murphy,
We would like to thank the reviewers for their time and thoughtful comments. Based on the reviewers’ comments, we have made edits to the figures, the introduction, methods, and discussion sections of the manuscript and believe we have a stronger article because of these suggestions.
Below we have detailed our responses to each of the reviewer comments. The reviewer comments are listed in Italics, and our responses are listed in red. Where applicable we have also listed the page, and line numbers corresponding to the changes in the manuscript. The changes in the manuscript have also been documented using tracked changes. We have submitted a tracked changes version, changes in bold version and a clean version of the revised manuscript.
Sincerely,
Sarvenaz Heirani Moghaddam
Reviewer 2
I have carefully reviewed your manuscript and would like to offer a few minor comments for your consideration.
Response: Thank to the reviewer for their suggestion and comments.
- Remove references from the abstract (e.g., see Fitts, 1954).
Response: Thank you for the suggestion, we have now removed this reference from the abstract.
- Introduction: It is not clear the rational for including individuals with Spinal Muscle Atrophy (SMA).
Response: Thank you for the suggestion to clarify the inclusion of our participant group. Participants with SMA were included because this population has the rare combination of largely intact central processing networks, little to no previous motor experience with upper-limb movements, and limited motor capabilities from birth. This population also represents an interesting contrast to the cervical SCI population, and stroke population used in previous studies (e.g., Eskanazi, 2009; Manson et al., 2014). We have amended our Introduction to clarify the rationale for the inclusion of people with SMA. Specifically, on page 3, line 96-107 we say:
“The goal of the present study was to examine if prior motor experience is critical for the action simulation process used to predict the other’s actions. To address this question, we selected participants with spinal muscle atrophy (SMA). SMA is a genetic neuromuscular disease that primarily affects motor neurons in the spinal cord leading to muscle loss and weakness in the upper and lower-limbs (see: [11]). Symptoms appear from 4-18 months of age and last throughout adulthood. Importantly, because of the early onset and nature of the pathology, people with SMA have largely intact central processing networks, but severely limited motor experience from birth.”
- The table: The demographic data obtained for each participant in each group should be moved in to the result section.
Response: Thank you for this suggestion. We have now moved Table 1 to the results section.
- Authors should discuss that the perception of action possibility judgments could be influenced by factors other than motor experience, such as cognitive processing differences or familiarity with the task. These factors might not have been adequately controlled for.
Response: Thank you for bringing up this additional point of discussion. We indeed agree with the reviewer that other factors could explain differences in perception, especially among people with SMA and we have added a section in the discussion (page 15 and lines 430-442) to address this. Specifically, considering the comments of reviewer 3, we have included recent literature about the impact of knowledge of task performance, and points from a review paper about the relationship between action observation, motor imagery and perception.
The section we have included reads:
“It is also important to mention that the perception of action possibility judgements can be influenced by factors beyond previous motor experience. Differences in cognitive processing and/or task familiarity could have contributed to the observed differences in predicted MTs [30,31]. For example in Zelaznik and Forney [32], found that perception of Fitts’s task difficulty (target width) was only related to motor performance if participants received a score after they performed the task. This finding indicates that knowledge of performance, in addition to motor experience is important for the accuracy of perceptual judgements (also see [33]). Furthermore, in the current study, it is possible that familiarity with speed accuracy tradeoffs could impact perceptual judgements. Although no studies have investigated whether repeated observation can improve perception in the absence of motor experience for upper-limb reaching tasks, it is possible that repeated action observation could improve prediction ability. This hypothesis is based on findings that action-observation and motor imagery facilitate motor skill acquisition (for a review see [34]).”
Reviewer 3 Report
After reviewing the manuscript, I can conclude that the authors have done a lot of work, the material is presented well, logical. The problem being developed is very interesting.
It is necessary to make changes to the title of the article, because in my opinion the use of the term "limited previous motor experience" is not entirely correct, it is possible to replace it with "limited motor capabilities", since all participants somehow had driving experience? If not, it can indicate this in the methods when describing the participants. And the word "evaluation" may have been omitted in the title, since it is not about one's own movement, but the evaluation of another person's action. Then the name will correspond to the presented problem and reflect the main message of the study.
The abstract fully reflects the results of the study. However, the annotations usually do not include references to literature. And the volume of the abstract according to the rules of the journal is no more than 200 words.
Introduction. The introduction needs to be finalized. This is due to the fact that the relevance of the research is not obvious from the presented text. Firstly, due to the use of "not new" sources, not a single one in the last 5 years, which already calls into question the relevance. And secondly, a detailed analysis of your own research with the nomination of two hypotheses is not entirely correct (you need to directly indicate in our study). It is necessary to attract literature on the possible influence of motor experience on the assessment of the movement of other people or their own, as well as that previous motor experience is crucial for the process of imitation of action (this, in my opinion, is another problem).
The goal needs to be rewritten because ".... to challenge the idea that..." cannot be the goal. This may be a hypothesis or more appropriate in the discussion. Moreover, it does not follow at all from the presented introduction that this problem exists.
Possible literature:
Zelaznik, H. N., & Forney, L. A. (2016). Action-specific judgment, not perception: Fitts' law performance is related to estimates of target width only when participants are given a performance score. Attention, perception & psychophysics, 78(6), 1744–1754. https://doi.org/10.3758/s13414-016-1132-5
Bian T, Wolpert DM, Jiang ZP. Model-Free Robust Optimal Feedback Mechanisms of Biological Motor Control. Neural Comput. 2020 Mar;32(3):562-595. doi: 10.1162/neco_a_01260. Epub 2020 Jan 17. PMID: 31951794.
The methods are described in detail and correspond to the tasks set. The selection criteria of the sample are clearly explained and justified. The participants are adequately described. The sample size is adequately represented, statistical processing has been carried out. Regarding ethical standards, you should write according to the requirements of the journal: When reporting on research that involves human subjects, human material, human tissues, or human data, authors must declare that the investigations were carried out following the rules of the Declaration of Helsinki of 1975 (https://www.wma.net/what-we-do/medical-ethics/declaration-of-helsinki/), revised in 2013. According to point 23 of this declaration, an approval from the local institutional review board (IRB) or other appropriate ethics committee must be obtained before undertaking the research to confirm the study meets national and international guidelines. As a minimum, a statement including the project identification code, date of approval, and name of the ethics committee or institutional review board must be stated in Section ‘Institutional Review Board Statement’ of the article.
Example of an ethical statement: "All subjects gave their informed consent for inclusion before they participated in the study. The study was conducted in accordance with the Declaration of Helsinki, and the protocol was approved by the Ethics Committee of XXX (Project identification code)."
The results of the study are statistically processed and presented in figures. The reliability is beyond doubt, the results are really important to confirm the hypothesis put forward.
The discussion requires the involvement of modern literature. And expanding the idea of the influence of limited previous motor experience on the perception of other people's actions. For example, to involve the concept of internal models in sensorimotor integration.
Tin, C., & Poon, C. S. (2005). Internal models in sensorimotor integration: perspectives from adaptive control theory. Journal of neural engineering, 2(3), S147–S163. https://doi.org/10.1088/1741-2560/2/3/S01
Egger, S. W., Remington, E. D., Chang, C. J., & Jazayeri, M. (2019). Internal models of sensorimotor integration regulate cortical dynamics. Nature neuroscience, 22(11), 1871–1882. https://doi.org/10.1038/s41593-019-0500-6
Ritz H, Frömer R, Shenhav A. Bridging Motor and Cognitive Control: It's About Time! Trends Cogn Sci. 2020 Jan;24(1):6-8. doi: 10.1016/j.tics.2019.11.005. Epub 2019 Nov 25. PMID: 31780248; PMCID: PMC6989175.
Technical remarks:
In the introduction, the formulas must be presented according to the rules, lines 49, 52. The article template:
https://www.mdpi.com/files/word-templates/brainsci-template.dot
According to the figures: the caption to the figure indicates "ID 1, 2 and 3.....", in all figures on the X–axis - 2, 3,4.
In Figure 2 – specify the color designations (red, blue). And change the stroke size so that it becomes noticeable in the 95% confidence interval designation. You should also increase the font size in the upper line, where the confidence and * parameters are given.
When writing the results, "±" should be indicated instead of "+". The spelling M and SD are usually given as M (SD).
According to the list of references: there is little modern literature. For example: [18], similar studies can be picked up over the past 5 years.
Brandone, A. C., Stout, W., & Moty, K. (2020). Intentional action processing across the transition to crawling: Does the experience of self-locomotion impact infants' understanding of intentional actions? Infant behavior & development, 60, 101470. https://doi.org/10.1016/j.infbeh.2020.101470
Or [22, 23], for example, Galli, J., Garofalo, G., Brunetti, S., Loi, E., Portesi, M., Pelizzari, G., Rossi, A., Fazzi, E., & Buccino, G. (2022). Children with Cerebral Palsy can imagine actions like their normally developed peers. Frontiers in neurology, 13, 951152. https://doi.org/10.3389/fneur.2022.951152
Xie, J., Jiang, L., Li, Y., Chen, B., Li, F., Jiang, Y., Gao, D., Deng, L., Lv, X., Ma, X., Yin, G., Yao, D., & Xu, P. (2021). Rehabilitation of motor function in children with cerebral palsy based on motor imagery. Cognitive neurodynamics, 15(6), 939–948. https://doi.org/10.1007/s11571-021-09672-3.
Line 56 missing link [2]
Line 71 missing link [1]
The design of the list of references should be according to the rules of the journal.
https://www.mdpi.com/files/word-templates/brainsci-template.dot
In conclusion, the study topic is very interesting and undoubtedly relevant, and the authors are respected researchers in the field. But, I think that publication can be considered only after revision.
Author Response
Dear Dr. Murphy,
We would like to thank the reviewers for their time and thoughtful comments. Based on the reviewers’ comments, we have made edits to the figures, the introduction, methods, and discussion sections of the manuscript and believe we have a stronger article because of these suggestions.
Below we have detailed our responses to each of the reviewer comments. The reviewer comments are listed in Italics, and our responses are listed in red. Where applicable we have also listed the page, and line numbers corresponding to the changes in the manuscript. The changes in the manuscript have also been documented using tracked changes. We have submitted a tracked changes version, changes in bold version and a clean version of the revised manuscript.
Sincerely,
Sarvenaz Heirani Moghaddam
Reviewer 3
After reviewing the manuscript, I can conclude that the authors have done a lot of work, the material is presented well, logical. The problem being developed is very interesting.
Response: Thank you to the reviewers for their supportive comments.
- It is necessary to make changes to the title of the article, because in my opinion the use of the term "limited previous motor experience" is not entirely correct, it is possible to replace it with "limited motor capabilities", since all participants somehow had driving experience? If not, it can indicate this in the methods when describing the participants. And the word "evaluation" may have been omitted in the title, since it is not about one's own movement, but the evaluation of another person's action. Then the name will correspond to the presented problem and reflect the main message of the study.
Response: Thank you for this comment and for the suggestion. We have now changed the title so that it reads: The impact of limited previous motor experience on action possibility judgements in people with spinal muscle atrophy. We added the word “judgement” but did not replace the “motor experience” with “capability” as motor experience is central to the research question presented in the study.
- The abstract fully reflects the results of the study. However, the annotations usually do not include references to literature. And the volume of the abstract according to the rules of the journal is no more than 200 words.
Response: Thank you for the suggestion. We have now edited the abstract (word count = 195) and removed all references.
- The introduction needs to be finalized. This is due to the fact that the relevance of the research is not obvious from the presented text. Firstly, due to the use of "not new" sources, not a single one in the last 5 years, which already calls into question the relevance. And secondly, a detailed analysis of your own research with the nomination of two hypotheses is not entirely correct (you need to directly indicate in our study). It is necessary to attract literature on the possible influence of motor experience on the assessment of the movement of other people or their own, as well as that previous motor experience is crucial for the process of imitation of action (this, in my opinion, is another problem).
Response: Thank you for the comment, we have updated the references and modified the introduction. For specifics on modifications to the introduction, see response to comment # 4 (next comment). To address the latter point we have used some of the references you have suggested in the discussion section where we addressed other contributions to action possibility judgements (e.g., cognitive processing and task familiarity).
- The goal needs to be rewritten because ".... to challenge the idea that..." cannot be the goal. This may be a hypothesis or more appropriate in the discussion. Moreover, it does not follow at all from the presented introduction that this problem exists.
Response: Thank you very much for the comment and suggestion. Together with comment #2 from Reviewer 2, we have revised the research question to better describe our purpose. Please see page 3 and line 96-107.
Possible literature:
Zelaznik, H. N., & Forney, L. A. (2016). Action-specific judgment, not perception: Fitts' law performance is related to estimates of target width only when participants are given a performance score. Attention, perception & psychophysics, 78(6), 1744–1754. https://doi.org/10.3758/s13414-016-1132-5
Bian T, Wolpert DM, Jiang ZP. Model-Free Robust Optimal Feedback Mechanisms of Biological Motor Control. Neural Comput. 2020 Mar;32(3):562-595. doi: 10.1162/neco_a_01260. Epub 2020 Jan 17. PMID: 31951794.
- The methods are described in detail and correspond to the tasks set. The selection criteria of the sample are clearly explained and justified. The participants are adequately described. The sample size is adequately represented, statistical processing has been carried out. Regarding ethical standards, you should write according to the requirements of the journal: When reporting on research that involves human subjects, human material, human tissues, or human data, authors must declare that the investigations were carried out following the rules of the Declaration of Helsinki of 1975 (https://www.wma.net/what-we-do/medical-ethics/declaration-of-helsinki/), revised in 2013. According to point 23 of this declaration, an approval from the local institutional review board (IRB) or other appropriate ethics committee must be obtained before undertaking the research to confirm the study meets national and international guidelines. As a minimum, a statement including the project identification code, date of approval, and name of the ethics committee or institutional review board must be stated in Section ‘Institutional Review Board Statement’ of the article. Example of an ethical statement: "All subjects gave their informed consent for inclusion before they participated in the study. The study was conducted in accordance with the Declaration of Helsinki, and the protocol was approved by the Ethics Committee of XXX (Project identification code)."
Response: Thank you for this important suggestion. We have now modified our ethics statement to match the journal rules and added the Project identification code.
- The results of the study are statistically processed and presented in figures. The reliability is beyond doubt, the results are really important to confirm the hypothesis put forward.
Response: Thank you for your supportive comments.
- The discussion requires the involvement of modern literature. And expanding the idea of the influence of limited previous motor experience on the perception of other people's actions. For example, to involve the concept of internal models in sensorimotor integration.
Tin, C., & Poon, C. S. (2005). Internal models in sensorimotor integration: perspectives from adaptive control theory. Journal of neural engineering, 2(3), S147–S163. https://doi.org/10.1088/1741-2560/2/3/S01
Egger, S. W., Remington, E. D., Chang, C. J., & Jazayeri, M. (2019). Internal models of sensorimotor integration regulate cortical dynamics. Nature neuroscience, 22(11), 1871–1882. https://doi.org/10.1038/s41593-019-0500-6
Ritz H, Frömer R, Shenhav A. Bridging Motor and Cognitive Control: It's About Time! Trends Cogn Sci. 2020 Jan;24(1):6-8. doi: 10.1016/j.tics.2019.11.005. Epub 2019 Nov 25. PMID: 31780248; PMCID: PMC6989175.
Response: Thank you very much for the comment and suggestion. Together with comment #4 from Reviewer 2, we have expanded on other factors that contribute to action possibility judgements to our discussion and included the relevant literature.
Technical remarks:
- In the introduction, the formulas must be presented according to the rules, lines 49, 52. The article template: https://www.mdpi.com/files/word-templates/brainsci-template.dot
Response: Thank you for this comment, we have now changed the format of how the equations are presented so that it matches the rules of the journal.
- According to the figures: the caption to the figure indicates "ID 1, 2 and 3.....", in all figures on the X–axis - 2, 3,4.
Response: Thank you for this comment, we have now fixed this issue.
- In Figure 2 – specify the color designations (red, blue). And change the stroke size so that it becomes noticeable in the 95% confidence interval designation. You should also increase the font size in the upper line, where the confidence and * parameters are given.
Response: Thank you for this comment. We have now modified figure 2 based on your suggestion.
- When writing the results, "±" should be indicated instead of "+". The spelling M and SD are usually given as M (SD).
Response: Thank you for this comment. We have reported all of the data according to the rules of the journal.
- According to the list of references: there is little modern literature. For example: [18], similar studies can be picked up over the past 5 years.
Brandone, A. C., Stout, W., & Moty, K. (2020). Intentional action processing across the transition to crawling: Does the experience of self-locomotion impact infants' understanding of intentional actions? Infant behavior & development, 60, 101470. https://doi.org/10.1016/j.infbeh.2020.101470
Or [22, 23], for example, Galli, J., Garofalo, G., Brunetti, S., Loi, E., Portesi, M., Pelizzari, G., Rossi, A., Fazzi, E., & Buccino, G. (2022). Children with Cerebral Palsy can imagine actions like their normally developed peers. Frontiers in neurology, 13, 951152. https://doi.org/10.3389/fneur.2022.951152
Xie, J., Jiang, L., Li, Y., Chen, B., Li, F., Jiang, Y., Gao, D., Deng, L., Lv, X., Ma, X., Yin, G., Yao, D., & Xu, P. (2021). Rehabilitation of motor function in children with cerebral palsy based on motor imagery. Cognitive neurodynamics, 15(6), 939–948. https://doi.org/10.1007/s11571-021-09672-3.
Response: Thank you for this comment. We have added more recent relevant literature to the manuscript based on your suggestion.
- Line 56 missing link [2]; Line 71 missing link [1]
Response: Thank you for bringing this up. We have now fixed both of these links.
- The design of the list of references should be according to the rules of the journal. https://www.mdpi.com/files/word-templates/brainsci-template.dot
Response: Thank you for pointing this out. We have now fixed this issue and are using the referencing style (Zotero: Multidisciplinary Digital Publishing Institute) accepted by the journal.
In conclusion, the study topic is very interesting and undoubtedly relevant, and the authors are respected researchers in the field. But, I think that publication can be considered only after revision.
Response: Thank you very much for your supportive and thoughtful comments.
Reviewer 4 Report
In this paper authors examined the impact of limited previous motor experience on action possibility judgements for others. Paper was well written and I suggest that this paper be accepted
Please review paper for minor language modifications.
Author Response
Dear Dr. Murphy,
We would like to thank the reviewers for their time and thoughtful comments. Based on the reviewers’ comments, we have made edits to the figures, the introduction, methods, and discussion sections of the manuscript and believe we have a stronger article because of these suggestions.
Below we have detailed our responses to each of the reviewer comments. The reviewer comments are listed in Italics, and our responses are listed in red. Where applicable we have also listed the page, and line numbers corresponding to the changes in the manuscript. The changes in the manuscript have also been documented using tracked changes. We have submitted a tracked changes version, changes in bold version and a clean version of the revised manuscript.
Sincerely,
Sarvenaz Heirani Moghaddam
Reviewer 4
In this paper authors examined the impact of limited previous motor experience on action possibility judgements for others. Paper was well written and I suggest that this paper be accepted
Response: Thank you very much for your supportive and thoughtful comments.